# Possibilities in Recycling Magnetic Materials in Applications of Polymer-Bonded Magnets

Uta Rösel * and Dietmar Drummer

Institute of Polymer Technology, Friedrich-Alexander-Universität Erlangen-Nürnberg, 91058 Erlangen, Germany; dietmar.drummer@fau.de
* Correspondence: uta.ur.roesel@fau.de

**Abstract:** Polymer-bonded magnets have increased significantly in the application of drive technology, especially in terms of new concepts for the magnetic excitation of synchronous or direct current (DC) machines. To satisfy the increasing demand of hard magnetic filler particles and especially rare earth materials in polymer-bonded magnets, different strategies are possible. In addition to the reduction in products or the substitution of filler materials, the recycling of polymer-bonded magnets is possible. Different strategies have to be distinguished in terms of the target functions such as the recovery of the matrix material, the filler or both materials. In terms of polymer-bonded magnets, the filler material—especially regarding rare earth materials—is important for the recycling strategy due to the limited resource and high costs. This paper illustrates two different recycling strategies relative to the matrix system of polymer-bonded magnets. For thermoset-based magnets, a thermal strategy is portrayed which leads to similar magnetic properties in terms of the appropriated atmosphere and process management. The mechanical reusage of shreds is analyzed for thermoplastic-based magnets. The magnetic properties are reduced by about 20% and there is a change in the flow conditions and with that, an influence on the pole accuracy.

**Keywords:** hard magnetic filler; flow and curing behavior; highly filled thermosets; polymer-bonded magnets

## 1. Introduction

Typical fields for polymer-bonded magnets are sensor and drive technology. In sensor systems, polymer-bonded magnets are normally a signal transducer, for example to detect linear and rotatory motion and position in combination with a Hall sensor [1] or a magnetoresistive sensor [2]. Especially in the automotive industry, many applications in sensor systems can be found, for example controlling the motion of the wiper or sunroof and register the torque and rotation angle for steering systems [3]. The main application within the drive technology is the magnetic excitation of synchronous or direct current (DC) machines [2]. Those applications have as of yet mainly been realized by thermoplastic-based polymer-bonded magnets as this type of magnets can be fabricated by injection molding and with that, the processing of multipolar and complex magnet field structures is suitable for series production.

Polymer-bonded magnets consist mainly of a matrix material and a filler, and in some cases of additives to improve the flowability or the adherence between the filler and matrix material. With respect to the matrix material, the typical fabrication method can be distinguished and further, the total filler content is defined. With that, injection-molded magnets are mainly based on a thermoplastic matrix and have only a filler content of about 60% of the volume [1]. Thermoset-based polymer-bonded magnets are primarily manufactured in a pressing process [4], where the filler content can reach up to 85% of the volume. With that, higher magnetic properties can be reached with respect to the injection-molded magnets [5]. The filler content determines the maximum of the magnetic properties a sample can have.

## 1.1. Magnetic Properties

Magnetic properties are based on the smallest magnetic unit within a solid and can be described by the magnetic moment of a single electron and its rotation around its axis, also named spin [6]. Hard magnets are part of ferromagnetic materials and have a characteristically high resistance against demagnetization [7]. The two main categories of the hard magnetic fillers are rare earth materials such as neodymium–iron–boron (NdFeB) and ceramic materials such as strontium–ferrite–oxide (SrFeO) [8]. The filler material has the main impact on the magnetic properties and reveals different magnetic properties, geometries and particle sizes. NdFeB particles exhibit a plate-like structure with a particle size of 100–400 μm and SrFeO a hexagonal geometry with a particle size of 1–10 μm. The resistance against demagnetization of NdFeB is two to three times higher than the resistance of SrFeO [9]. The filler material can have isotropic or anisotropic magnetic properties [10]. As the magnetic moments in isotropic filler particles are orientated randomly, samples do not have a preferential direction regarding the magnetic properties. Due to the preferential direction of the magnetic properties and the crucial orientation of anisotropic fillers in the process, samples with anisotropic fillers reach a remanence $B_R$ of about 85% of the saturation flux density $B_S$, depending on the filler content and the quality of the production process. The remanence $B_R$ of samples with isotropic fillers obtains only 50% of BS [11]. The orientation of fillers requires that the fillers be mobile and exhibit a magnetic moment [11]. The magnetic moments of the anisotropic filler particles have to be oriented and magnetized within the production. The magnetization can also be done afterwards, for example using an impulse magnetization [12].

## 1.2. Urgent Requirement of Recycling Strategies for Polymer-Bonded Magnets

The focus of the recycling strategies is mainly on rare earth materials in terms of polymer-bonded magnets as there are limited resources of them. NdFeB is the third leading rare earth material and in terms of permanent magnets the strongest one so far [13]. A total of 90% of the worldwide production of NdFeB are used to fabricate permanent magnets [14]. In 2012, 59% and in 2018 even 89% of the worldwide utilized NdFeB were applied in drive technologies as electric motors in the industry, e-mobility and wind energy. In 2030, it is assumed that these applications will consume 95% of the resource, mainly due to the fact that electromobility is highly encouraged to be standardized as the alternative drive technology and further, new motor concepts with integrated magnets are explored [15]. However, within these applications only 6% were fabricated by polymer-bonded magnets, due to the fact that so far, the technology for manufacturing polymer-bonded magnets in driving applications is nearly unknown [15]. The first investigations for these applications based on thermoplastics show the possibility of implementing polymer-bonded magnets by injection molding in synchronous reluctance motors, with respect to limited chemical permanence and thermal resistance [16]. Nevertheless, NdFeB—representative for rare earth materials—is highly used in permanent magnets. The implementation of new concepts in driving technology based on polymer-bonded magnets is a major factor in realizing the actual demand in the applications due to multipolar and complex magnetic field structures. Therefore, the recycling strategies have to be focused on these upcoming applications in terms of polymer-bonded magnets. It is assumed that the percentage of them in terms of the usage of NdFeB will increase tremendously in the next 10 to 20 years. Between 2014 and 2020, the demand for NdFeB magnets was doubled up to about 120,000 t [17]. Due to this massive increase of the use of the limited resource NdFeB, it is essential to realize recycling strategies for polymer-bonded magnets to satisfy the increasing demand. With respect to [17], Germany used about 1000 t of NdFeB within industrial driving applications in 2014. In 2014, 35 t of this stock could be utilized for recycling. It is presumed that in 2030, 100 t of the stock will be available for reusage. This is because most of the industrial driving applications reveal a durability of 10 to 15 years, leading to longer return times. Further, a large number of returns is sold to foreign countries [17]. However, the increasing demand for rare earth magnets and the upcoming awareness of resource-saving consuming requires

the exploring and successful establishment of recycling strategies of polymer-bonded magnets based on rare earth materials. The opportunity to recycle will lead to the possibility of increasing returns.

### 1.3. Reduction and Recycling Possibilities of Polymer-Bonded Magnets

To satisfy the increasing demand of rare earth materials in magnets, different strategies are possible. The reduction of rare earth materials in products can be realized by efficiency enhancements leading to less material usage in products together with a simultaneous increase of the output [15]. This can be realized by implementing new fabrication strategies of polymer-bonded magnets, for example in rotor applications, to increase the usage of the geometry freedom of the injection-molding process. The first investigations based on thermoplastics have revealed the increase of the inner torque by 33% and the reduction of the component size by up to 70%, leading to a high reduction in the use of the material [18].

Another strategy is the substitution of rare earth materials where three groups can be distinguished: substantial (chemical elements are exchanged, for example neodymium for praseodymium), material (different material system with similar properties) or functional (new technology) substitution [15]. The first two strategies barely solve the problem of the limitation of rare earth materials as the substantial substitution shifts the problem to a different rare earth material and the material substitution is not able to reach the magnetic properties of rare earths yet. Magnetic fillers based on ceramics exhibit only one third of the magnetic properties of rare earths [19]. With that, the functional substitution is currently the only group with a promising potential of success.

The third strategy is the recycling of polymer-bonded magnets, where different target functions have to be distinguished. The recovery of the matrix material, the filler or both materials can be the focus of the strategy. In terms of polymer-bonded magnets, the filler material—especially regarding rare earth materials—is important for the recycling strategy due to their limited resource and high costs and is therefore the focus of further investigations. The successful implementation of recycling strategies requires the establishment of a closed-loop economy where possibilities of return and disassembly are given. With that, the investigations of recycling strategies for polymer-bonded magnets go along with the establishment of collection and segmentation systems [17].

The recycling strategies can further be distinguished into three groups of different treatments: thermal, chemical and mechanical. Figure 1 depicts these groups with different realization possibilities and the potential of implementation on thermoplastics or thermosets. The different strategies are mainly applied to unfilled polymers or filled polymers without the focus on recovering the filler.

Standard recycling strategies for thermosets are the energy recovery and the reuse of shreds. This implementation of infusible particles in the process is not sufficient in terms of polymer-bonded magnets, as these parts act as foreign parts without the ability to orient. The chemical strategies are subjects of current research. In [20], a new method of solvolysis for thermosets was presented in terms of an epoxy resin, type Bisphenol F and the hardener diamino diphenyl methane. Using a nitric acidic solution at 85 °C, the C-H bonds were broken and the hardener was separated from the resin. The decomposed products were repolymerized with an original resin or curing agent, reaching sufficient mechanical properties [20]. The method can only be applied to material systems in which the suitable solution conditions relative to the thermoset are known. Further, a new material group of thermosets was found, which enables thermosets to reveal a reversible network—so-called covalent adapted network or dynamic covalent network [21]—by bond exchange reactions. In these special thermosets, an active unit without a bond attaches to an existing bond and builds a new ternary bond. This connection is unstable which leads to a fragmentation process and the generation of a new bond and a new active unit [22]. For example, ref. [23] found an epoxy resin where the bond exchange reaction is realized by transesterification reactions. However, again a special matrix material in terms of a polymer-bonded magnet was necessary to realize the recycling process. The third chemical route of

recycling thermosets requires the integration of specific thermoplastic or photosensitive monomers [24]. For example, the monomer C double bonds of diclycopentadiene (DCPD) were broken by a catalyst and a modified poly-DCPD was built by adding about 10 weight percent of a special linker (silyl ether). Fluoride ions were able to influence the linker in that way, such that the modified poly-DCPD networks separated itself into linker and DCPD monomers [25]. Again, this mechanism was only developed for specific materials. However, the physical basis of the method can be applied to any thermoset with respect to the fact that the precise combination of linker and monomer has to be found for each material system.

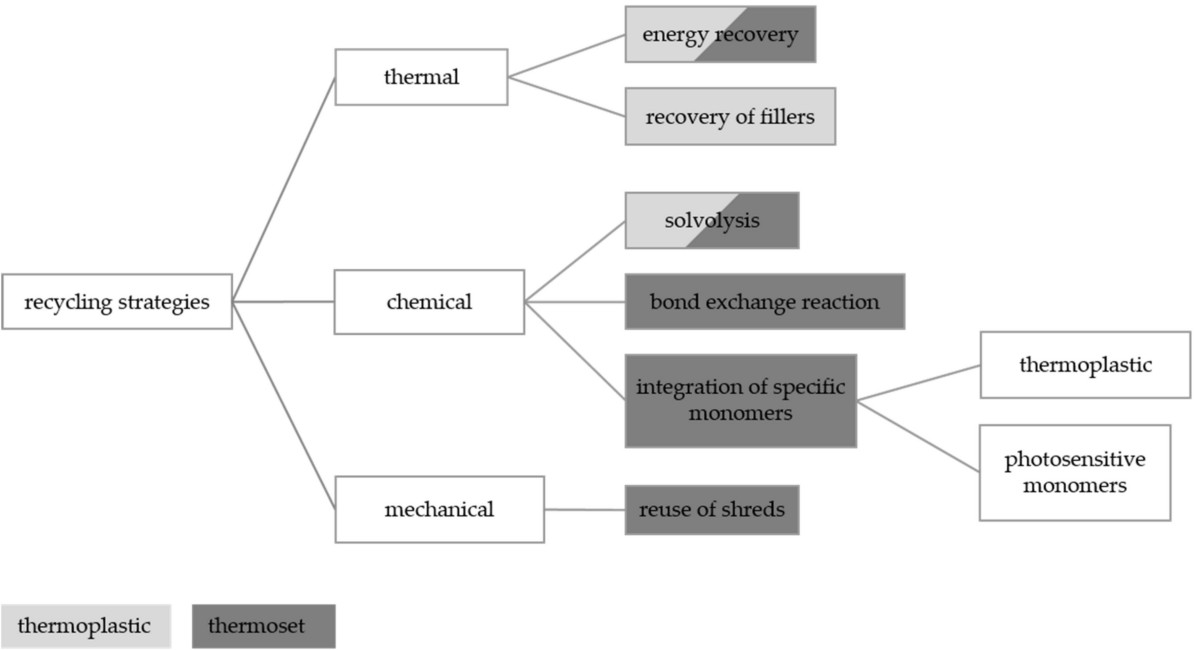

**Figure 1.** Overview of different recycling strategies and the potential of implementation on thermoplastics or thermosets.

The recycling of thermoplastics is mainly realized by fusing shreds again. This is a basic difference in the recycling strategies of thermoplastics and thermosets as the re-fusing is not possible in terms of thermosets. However, thermoplastics can age through multiple fusions. In terms of thermal recycling, thermoplastics can be used to recover energy or fillers as long as their thermal stability is higher than the decomposition temperature. Further, thermoplastics can be reduced in monomers by applying the solvolysis method.

With respect to the different recycling strategies and the possible implementation on thermoplastics and thermosets, a suitable method for polymer-bonded magnets has to be found. So far, recycling strategies for permanent magnets have been focused on sintered ones. In [26], different process routes for the reutilization of sintered permanent magnets in electromotors were investigated and a chemical route with a metallurgical treatment was identified to reach the best results. In [27], a method of recycling polymer-bonded magnets was described, but only in a theoretical way, by removing the matrix material through a solvent and reusing the extracted filler [28]. However, no investigations were done in terms of adopting this method in several applications. Therefore, there was no knowledge about the feasibility or the magnetic quality of the recycled magnets. Further, the successful reusage of the sintered permanent magnets of an electric bicycle in polymer-bonded magnets was proven [29]. Nevertheless, the investigations did not reveal a concrete solution for the recycling of polymer-bonded magnets, as the initial point was always sintered magnets.

The aim of this paper was the investigation of a thermal recycling strategy for thermosets in order to recover the filler material independently from the matrix material system

as it would be in terms of a chemical strategy so far. Further, this thermal method is compared to the mechanical reuse of shreds for thermoplastics with re-fusing the material to show the possibilities of both reasonable methods with respect to the matrix system and to establish a recycling strategy for polymer-bonded magnets.

## 2. Materials and Methods

### 2.1. Material

The experiments can be divided into two recycling methods, in which different material systems were investigated. As the focus of the paper was the thermal recycling strategy for thermosets, this strategy was realized using two different filler materials, which are commonly used in terms of polymer-bonded magnets. The mechanical reuse of thermoplastics was conducted with only one of these filler systems to enable a comparison of both strategies in terms of the two matrix systems.

For the thermoset the matrix material was an epoxy resin (EP) mixture named Epofix (Struers GmbH, Ottensoos, Germany). The two components of resin and hardener have to be mixed at the ratio of 25 to 3. The thermoplastic matrix material was Polyamide 12 (PA 12), type Vestamid BS 1636 (Evonik Industries AG, Essen, Germany).

The experiments were conducted with the hard magnetic particles of anisotropic SrFeO and the anisotropic NdFeB. In terms of the thermoplastic material, only SrFeO was considered. For SrFeO the anisotropic type was OP 71 (Dowa Holdings Co., Ltd., Tokyo, Japan) and for NdFeB the anisotropic type was MQA 38-14 (Magnetquench GmbH, Tübingen, Germany). Table 1 presents the mean particle size of the filler material regarding the manufacturer specifications, where SrFeO reveals a much lower mean particle size relative to NdFeB. Further, the density, the thermal conductivity and heat capacity of the filler and matrix material are shown in Table 1, as these factors are important properties with respect to the flow and curing process.

**Table 1.** Specification of filler and matrix material including mean particle size of filler (manufacturer specifications) and density, thermal conductivity and heat capacity.

| Filler Material (-) | Type in (-) | Mean Particle Size in μm | Density in g/cm$^3$ | Thermal Conductivity λ in W/(mK) | Heat Capacity c in J/(gK) |
|---|---|---|---|---|---|
| SrFeO | OP 71 | 1.25 | 5.382 | 2.30 | 0.639 |
| NdFeB | MQA 38-14 | 105 | 7.501 | 6.10 | 0.423 |
| EP | Epofix | - | 1.100 | 0.20 | 1.384 |
| PA12 | Vestamid BS 1636 | - | 1.009 | 0.33 | 1.840 |

Figure 2 depicts the geometry of the anisotropic filler material SrFeO (hexagonal) and NdFeB (plate-like) using a scanning electron microscope (Gemini Ultra-Plus; manufacturer: Carl Zeiss AG, Oberkochen, Germany). The difference of the two filler types is portrayed not only in terms of the particle size, but also in terms of the geometry and the tendency to agglomerate.

The two different fillers were chosen in terms of the investigation, as each of them represent an exponent of the two types of hard magnetic fillers. As these two types—the hard ferrites and the rare earths—depict a major difference in terms of their particle properties and their behavior during fabrication but reveal an essential role in terms of the application, these two different types of fillers were portrayed. The authors are aware of the fact that this leads to differences in the evaluation of the results, for example in terms of different scales in the microscopy images. However, as both filler groups are essential in terms of polymer-bonded magnets, both filler types were investigated. To compare the behavior of the filler in more detail, especially in terms of the different filler size, the particle size distribution

is shown in Figure 3 for numerical and volumetric measurements. The difference in the particle size can be especially seen in terms of the volumetric measurement.

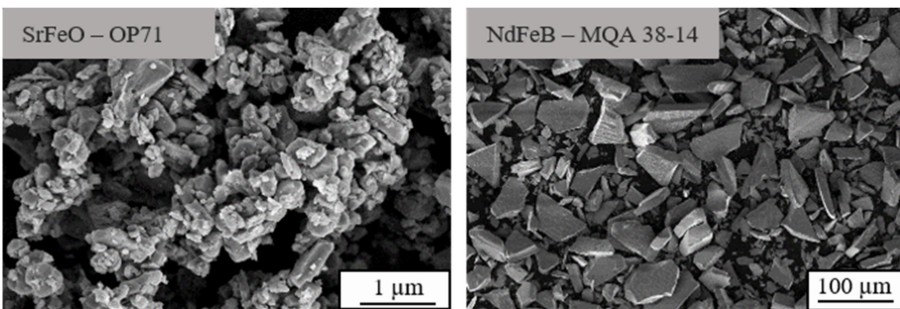

**Figure 2.** Particle geometry of anisotropic SrFeO and NdFeB using a scanning electron microscope.

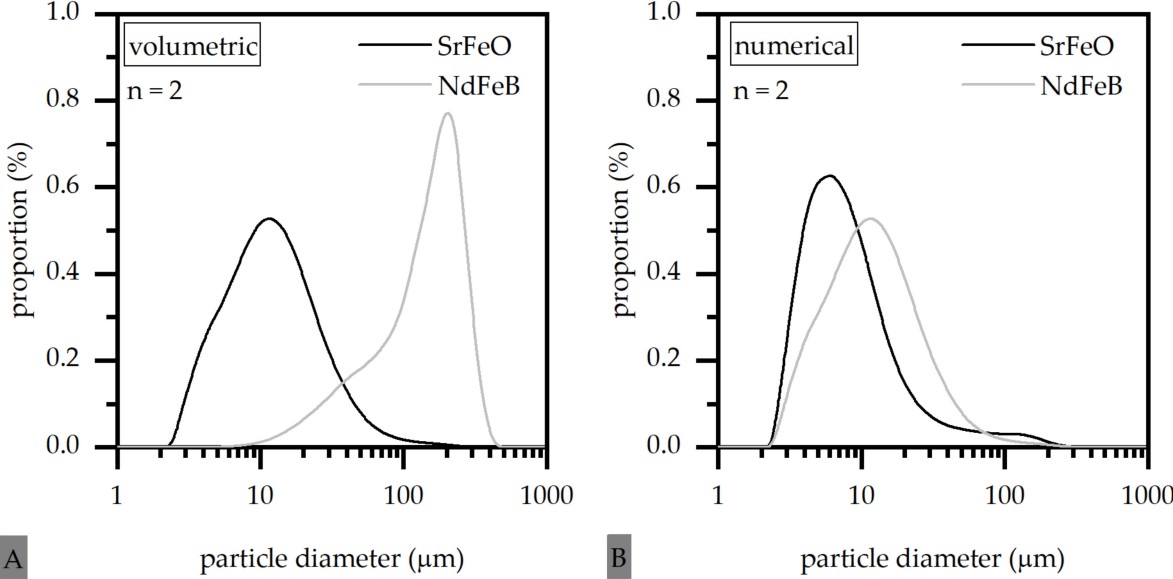

**Figure 3.** Mean particle size distribution comparison between anisotropic SrFeO and NdFeB based on volumetric (**A**) and numerical (**B**) measurements.

In terms of the thermoset matrix, the amount of the filler in the compound was varied between 40 and 70 vol.-% in intervals of 10, as these parameters reveal the lower limit with respect to relevant magnetic properties and the upper limit relative to a possible implementation of the highly filled fabrication of the compound. The thermoplastic-based compound was based on a 55 vol.-% filler content to ensure a good comparison between the limitations chosen in the thermoset compound and to generate a comparison between thermoset- and thermoplastic-based recycling strategies. As in thermoplastic-based compounds, the filler amount was limited to 60 vol.-% [30], and a slightly smaller amount with 55 vol.-% was investigated to reach sufficient magnetic properties with less particle–particle interactions due to the high filler content.

### 2.2. Fabrication of the Test Specimens
2.2.1. Thermoset-Based Test Specimens

The resin component of the matrix material was mixed with the filler with respect to the filler amount needed. Afterwards, the hardener was added, and the compound was placed in a crucible with a diameter of 24 mm. The curing process started after adding the hardener. With that, the filling of the crucible was limited in time to ensure that the particles had enough mobility to orientate. Underneath the crucible, a permanent magnet was placed to realize the orientation and partial magnetization of the fillers. The curing

process was finished after 8 h and the test samples were removed from the crucible. The curing time was quite long in terms of economic efficiency. However, the matrix material allowed the fabrication of test samples with only a little amount of filler. This was important in terms of the fabrication of test samples after the thermal recycling strategy. Ten samples were produced per setting to enable the production of recycled samples.

### 2.2.2. Thermoplastic-Based Test Specimens

The thermoplastic-based compounds were prepared in a twin-screw extruder, type ZSE HP-40D (Leistritz AG, Fürth, Germany). Both the filler and the polymer granules were added gravimetrically at different positions along the screw using a doser (K-Tron Deutschland GmbH, Genhausen, Germany). After the compounding, cooling was carried out by means of a vibratory feeder, followed by pelletizing to avoid contact with water. The compound revealed a density of 3.24 g/cm$^3$ with respect to the filler content.

The test samples were produced and the pressure controlled by a Demag Ergotech 25/280-80 injection-molding machine (Sumitomo (SHI) Demag Plastics Machinery GmbH, Schwaig bei Nuremberg, Germany) with a screw diameter of 18 mm. The processing parameters were set as shown in Table 2 with a 12-pole outer magnetic field. The circular test samples revealed an outer diameter of 30.6 mm and an inner diameter of 22.6 mm with a width of 5 mm. The four pinpoints of the gating system were placed in the middle of the pole on the inner side of the ring. With that, the weld line coincided with the pole crossover. The processing parameters had to be changed in terms of the fabrication of the shreds—mainly the holding pressure and the changeover point. Two hundred samples were produced to enable the production of recycled samples.

**Table 2.** Processing parameters of injection molding to fabricate test samples (PA12 with 55 vol.-% SrFeO).

| Mass temperature | 280 °C |
| --- | --- |
| Mold temperature | 80 °C |
| Injection speed | 80 mm/s |
| Changeover point | 1000 bar |
| Holding pressure | 500 bar |

### *2.3. Realization of Recycling Strategy*

### 2.3.1. Thermal Recycling Strategy for Thermoset-Based Polymer-Bonded Magnets

The samples were incinerated using a microwave oven (Phoenix Black—Cem GmbH, Kamp-Lintfort, Germany) and applying a temperature of 400 °C for 20 min followed by 600 °C for 50 min. During the process, the samples were placed in a ceramic crucible without and with a cover to change the impact of the atmosphere as shown in Figure 4A,B. Without a cover, the sample is less protected, and oxidation is likely to occur. Further, the pure filler materials were incinerated in the same way. The regained fillers of the samples after incineration were used to fabricate three new samples per setting based on the recycled fillers.

### 2.3.2. Mechanical Reuse of Shreds for Thermoplastic-Based Polymer-Bonded Magnets

After the ring samples were magnetized during the fabrication process, a mechanical reuse was only possible in case of a demagnetization of the samples. Otherwise, the granulation and fabrication of the recycled compound would not be possible. As the demagnetization is an expensive process, first attempts for the strategy were done using the gating system. The gating systems were regranulated in a mill and dried in a convection oven at 80 °C for 4 h. Afterwards, the compound was fabricated again using the injection-molding machine and the same parameters as shown in Section 2.2.2 and Table 2. Five samples were produced based on the mechanical reusage of shreds.

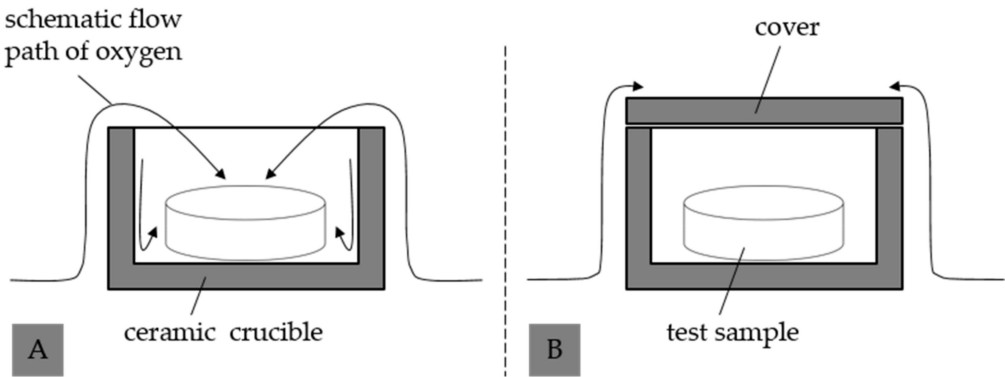

**Figure 4.** Schematic flow path of oxygen and interference with the test sample during the thermal recycling strategy without (**A**) and with (**B**) cover.

*2.4. Characterization*

2.4.1. Magnetic Properties

In the case of the thermoset-based samples, the magnetic property and more precisely the remanence $B_R$, was evaluated by measuring the hysteresis loop using a permagraph (type: C-300, Magnet-Physik Dr. Steingroever GmbH, Cologne, Germany). The permagraph applies a magnetic field strength H from outside onto the samples perpendicular to the circular interface, and a sensor gathers the magnetic flux density B relative to H. To realize a full magnetization of the filler particles, a pulse magnetizer (type: Im-12220-U-MA-C, Magnet-Physik Dr. Steingroever GmbH, Cologne, Germany) and a magnetic device (type: MV D30X30 mm F-TC, Magnet-Physik Dr. Steingroever GmbH, Cologne, Germany) were used before each measurement. Based on the remanence $B_R$, the degree of orientation $\delta$ was calculated with respect to Equation (1), where $B_{R,anisotrop}$ is the theoretical maximum of the remanence $B_R$ of the pure filler and $\varphi$ is the filler content. $B_{R,anisotrop}$ of NdFeB is 1500 mT and 450 mT in the case of SrFeO [19].

$$\delta = \frac{B_R}{B_{R,anisotrop} \cdot \varphi} \tag{1}$$

Further, the magnetic properties of the pure filler before and after incineration was measured using a cylindrical Hall sensor. The Hall sensor was placed perpendicular to a ceramic cavity in which the filler material was placed. The distance between the tip of the Hall sensor and the filler surface was kept constant with 0.5 mm to ensure that the conditions and especially the influence of air on the magnetic properties were the same throughout the evaluation.

For the thermoplastic-based ring samples, the magnetic flux density was determined using a test rig, as shown schematically in Figure 5. Here, the ring samples were picked up via a clamping device. The flux density was recorded radially via a Hall sensor relative to the rotation angle of the rotor. The angle of rotation was received using a rotary encoder from Heidenhain GmbH (Traunreut, Germany). For the radial data acquisition on the circumference of the samples, the shaft was driven by a motor with an adjustable speed and number of revolutions or running time. This made it possible to determine the course of the magnetic flux density for 360° or any multiple. The magnetic properties were analyzed on three characteristic positions of the ring: in the middle of the pole (B), on the left (A) and the right (C) of the weld line.

All measurements regarding the magnetic properties were conducted at room temperature.

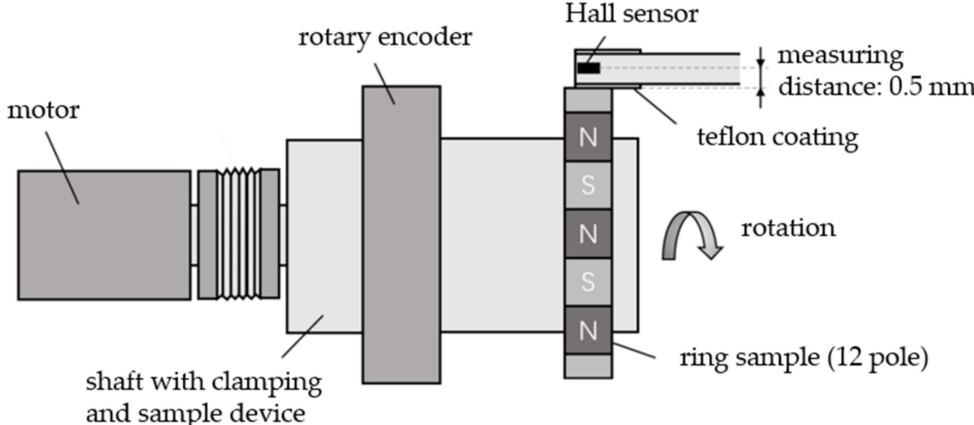

**Figure 5.** Schematic setup of the test rig for determining the magnetic flux density on the ring.

### 2.4.2. Filler Orientation

To analyze the filler orientation, the samples were embedded in cold-curing epoxy resin (type: Epofix, Struers GmbH, Ottensoos, Germany). The specimens were then split in the center using a water-cooled saw with minimal temperature input so that microscopic examinations could be performed in the center of the specimen and perpendicular to the expected long axis of the filler. The preparation of the two samples (out of the crucible and the ring) together with the expected orientation are shown in Figure 6. The specimens were further demagnetized and polished.

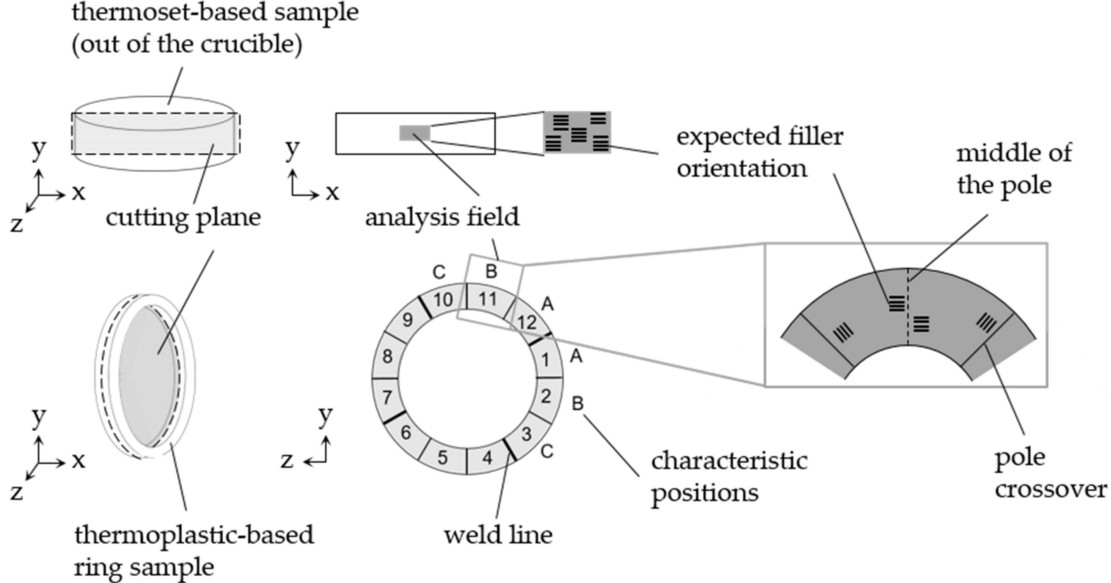

**Figure 6.** Sample preparation to determine the filler orientation with expected orientation.

In the case of NdFeB, the orientation was characterized by a stereo microscope (type: Axio Zoom.V16, Carl Zeiss AG, Oberkochen, Germany). As the mean particle size of SrFeO is significant lower relative to NdFeB, a scanning electron microscope (type: Gemini Ultra-Plus, Carl Zeiss AG, Oberkochen, Germany) was used to determine the orientation. Here, a 10 nm layer of spray gold was placed on top of the samples. On the basis of the images taken by the stereo or the scanning electron microscope, a differentiation between the matrix material and the filler was carried out by means of a grey scale threshold analysis. The orientation was evaluated along the longest axis of the individual particles in order to determine the main orientation angle between 0° and 180°. According to the formation of

the histogram, a preferred orientation of the fillers or a reduced orientation due to a broad scattering of the histogram could be deduced.

### 2.4.3. Thermogravimetric Analysis (TGA) Following DIN EN ISO 11358

To evaluate the change of the mass as an indicator of the oxidation of the sample during the incineration and to further evaluate the influence of the atmosphere on the magnetic properties of the recycled filler, a thermogravimetric analysis (TGA) was realized on the basis of the DIN EN ISO 11358 standard (type: TGA–Q 5000, TA Instruments, New Castle, Delaware, DE, USA). The thermoset-based compound with 40 vol.-% of filler amount as well as the pure filler was evaluated under nitrogen and oxygen atmosphere. The change of the mass was determined while the temperature of the sample was raised between 50 and 600 °C with a rate of 20 °C per minute.

### 2.4.4. Energy Dispersive X-ray Spectroscopy (EDX)

The energy dispersive X-ray spectroscopy is part of the material analyses and gives the opportunity to evaluate the amount and type of elements in the material. Here, the sample was stimulated on an atomic level by an electron beam. The returned X-rays were detected relative to the specific element. The pure filler before and after incineration with and without cover during the heat treatment in the ceramic crucible were characterized.

### 2.4.5. Differential Scanning Calorimetry (DSC) Following DIN EN ISO 11357

To characterize the heating and cooling behavior of the thermoplastic-based compound as well as the influence of the mechanical reuse of shreds, DSC measurements were carried out on the basis of the DIN EN ISO 11357 standard using pellets and a weight of about 9.8 mg on the DSC of TA-Instruments (New Castle, Delaware, USA). The material was tempered at 0 °C for 10 min before being heated to 220 °C at a rate of 20 °C per minute in the first heating cycle. After an isothermal hold of 0.5 min, the material was cooled to 0 °C at a rate of 10 °C per minute and held isothermal for 5 min before being reheated to 220 °C in the second heating cycle. The melting peak temperature during the second heating and the crystallization peak temperature during cooling were determined.

## 3. Results and Discussion

### 3.1. Magnetic Properties of Thermoset-Based Polymer-Bonded Magnets with Respect to the Thermal Recycling Strategy

The magnetic properties of the thermoset-based compounds for both filler types and a filler content between 40 and 70 vol.-% is shown in Figure 7A. Further, the degree of orientation is depicted in Figure 7B. The magnetic properties increase with rising filler content for both filler types. The properties reach a high level with 50 vol.-% NdFeB and hardly increase further up to 70 vol.-%. In the case of SrFeO, the magnetic properties increase more continuously. The degree of orientation is relatively low for both fillers. However, the degree is reduced in the case of SrFeO with increasing filler content and in the case of NdFeB with more than 50 vol.-%.

The magnetic properties of the samples as well as the degree of orientation reveal a certain degree of immobility of the filler particles as the degree of orientation is less than one and therefore, the remanence is smaller than the theoretical possible value in terms of the filler content in the sample. The mobility can be hindered by interactions between the particles or a fast-increasing viscosity of the matrix. However, these results are set as a reference point for the recycled samples to compare the change in the orientation and the magnetic properties.

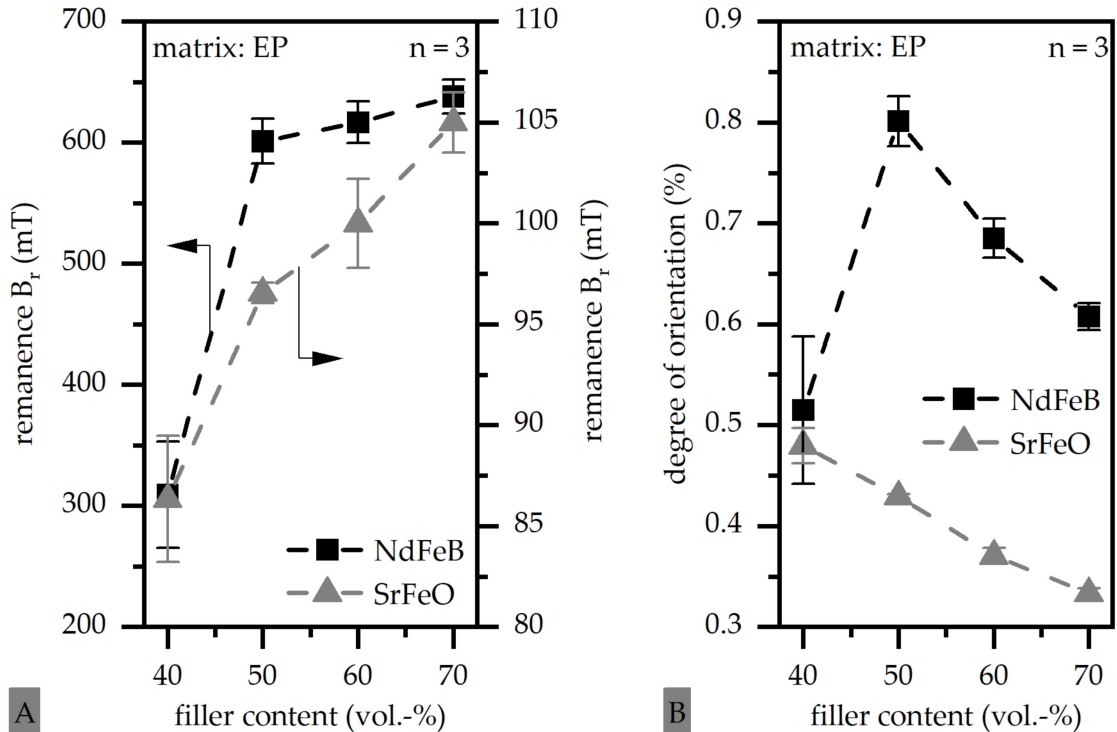

**Figure 7.** Magnetic properties (**A**) and degree of orientation (**B**) of thermoset-based compounds before the recycling step.

In Figure 8A, the remanence of the pure filler and the compound is shown in terms of the treatment and the atmosphere for both filler types. Due to the incineration without the cover, the remanence of filler and compound is reduced, whereas the value is similar for the samples without treatment and after incineration with a cover. Further, in Figure 8B, the remaining share and the reduction of the remanence is portrayed for both filler types. Here, the values between filler and compound as well as between the pure filler is compared. In terms of the comparison of filler and compound, the treatment is kept constant. For example, the issue "F1/C1" compares the value of the pure filler and the compound, both without treatment (1). It shows the reduction of the magnetic properties only based on the matrix added in the compound. The issue "F2/C2" compares the change of the magnetic properties in the compound relative to the filler, taking not only the factor of the matrix material into account but also the treatment (2). The issue "F1/F2" compares the pure filler in different stages of the treatment, or more precisely the reduction of the magnetic properties of the pure filler (F1) due to incineration without a cover (F2). The remaining share is significantly lower in terms of the comparison between filler and compound. The change of magnetic properties in terms of the pure filler is negligible when using a cover (Figure 8B, issue "F1/F3") and between 30 and 60% when not using a cover (Figure 8B, issue "F1/F3") during the incineration. In terms of SrFeO, the reduction in the pure filler is less compared to NdFeB (Figure 8, issue "F1/F2"), but higher in the compound (Figure 8, issue "F2/C2"). This reveals a reduction of the magnetic properties not only due to the oxygen atmosphere but, further, due to other disturbing influences such as particle–particle interactions and agglomerates.

The reduction of the magnetic properties is significantly increased by the oxygen atmosphere. Further, the remanence is reduced in the compound in comparison to the pure filler due to air gaps in terms of the samples without treatment, which were analyzed by using a scanning electron or stereo microscope. In the oxygen atmosphere, a further reduction of the magnetic properties takes place as agglomerates are built for SrFeO and an orientation in the flow direction occurs for NdFeB. Figure 9 supports this, where the orientation is shown for both filler types with 50 vol.-% of filler content in the sample

without treatment and after incineration without a cover. As the sample after incineration with a cover reveals similar magnetic properties to the sample without treatment, the orientation is not shown. The orientation of the sample without treatment is flat and as expected (compared to Figure 6). After incineration, the sample with SrFeO could not be analyzed due to the agglomerates and the sample with NdFeB shows a narrow orientation of 90°. This orientation is assumed to be reached due to the flow conditions, as the oxygen layer on the surface of the filler disables an orientation due to the outer magnetic field. The agglomerates in terms of SrFeO are likely to occur for the smaller particles of SrFeO compared to NdFeB. To clarify the building of agglomerates in terms of SrFeO and the orientation of NdFeB, Figure 9 further shows stereo microscope images of the samples after incineration without a cover.

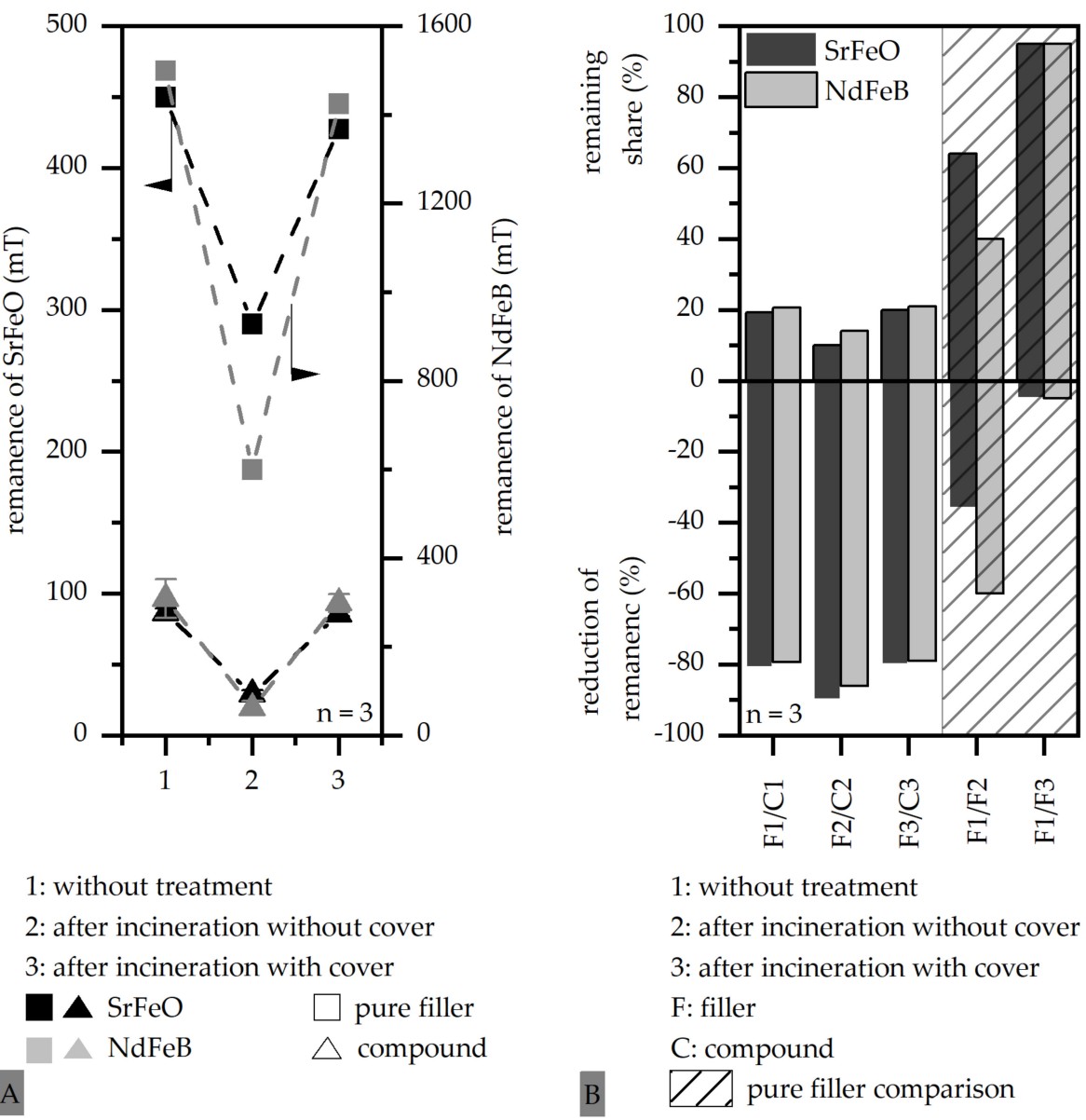

**Figure 8.** Magnetic properties of the pure filler and the compound (**A**) and comparison of magnetic properties between filler and compound as well as pure filler (**B**) in terms of the treatment.

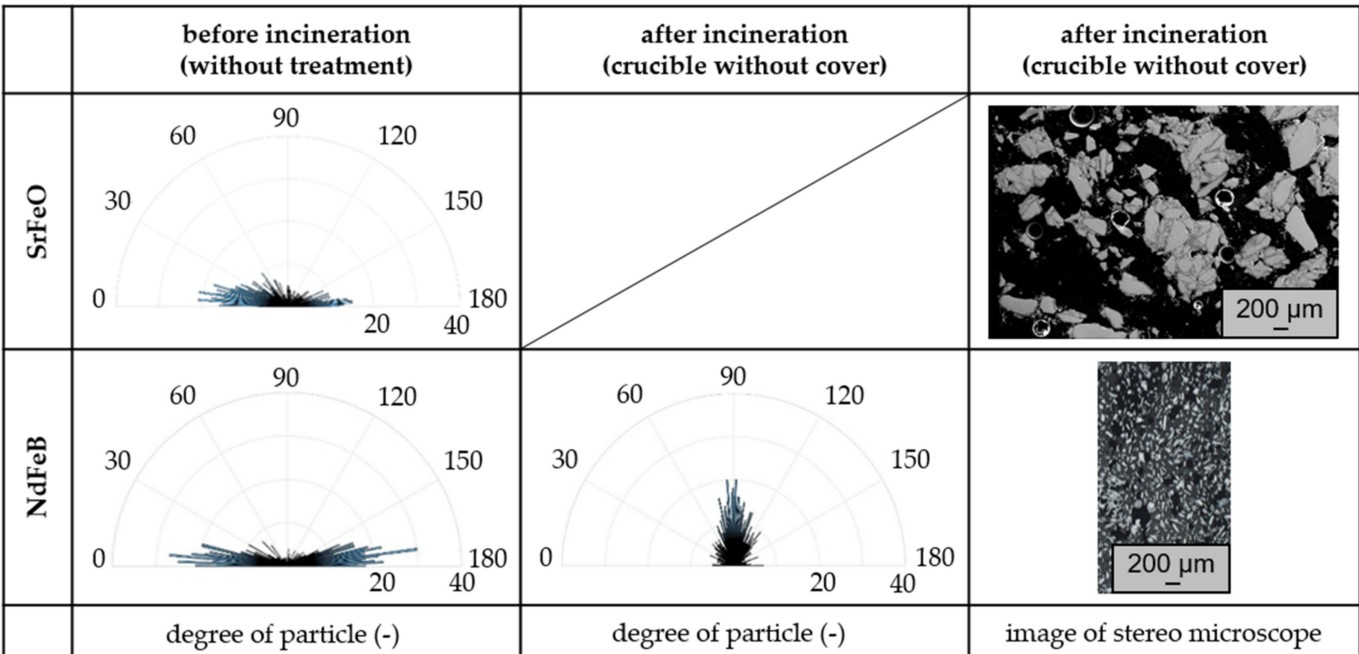

**Figure 9.** Filler orientation of samples with 50 vol.-% filler content before and after incineration with oxygen atmosphere.

*3.2. Influence of the Atmosphere and the Oxidation of Thermoset-Based Polymer-Bonded Magnets with Respect to the Thermal Recycling Strategy*

The change of the mass during the incineration is shown in Figure 10A for the pure filler NdFeB and SrFeO with oxygen and nitrogen atmosphere as well as for the thermoset compound with a 40 vol-.% filler content for both filler types and atmosphere types in Figure 10B. The mass of pure SrFeO does not change regardless of the atmosphere during incineration. However, the mass of NdFeB increases with a temperature greater than 480 °C under the oxygen atmosphere. The change of mass is reduced for both filler types in the compound, reaching a constant level at 480 °C for SrFeO in both atmospheres and for NdFeB in the nitrogen atmosphere.

The compound does not reach the exact amount of filler with respect to the given volumetric and analyzed weight percentage. This might occur due to unsteady filler content in the test sample. However, it can be clearly seen that the oxygen atmosphere has only a slightly negative effect onto the filler NdFeB in the compound. The difference in the change of the mass in both atmospheres is equivalent to about 40.23 vol.-% in the case of SrFeO (0.23 vol.-% deviation from nominal value) and about 43.3 vol.-% in the case of NdFeB (3.3 vol.-% deviation from nominal value). Further, a constant level even with the oxygen atmosphere is reached in the change of mass in the compound at 480 °C. The incineration could therefore be stopped at this temperature, leading to no harm in the NdFeB filler under an oxygen atmosphere. With respect to the thermogravimetric analysis, the incineration can be realized under an oxygen atmosphere up to 480 °C, as within this range, only a little oxidation takes place in terms of NdFeB. The expensive and complex integration of nitrogen in the process of the incineration is not necessary. However, a cover of the ceramic crucible is needed, as shown in Figure 4B.

The influence of the atmosphere can be further seen with the EDX spectroscopy. In Figure 11, the change of the composition of the elements in the pure filler NdFeB with oxygen atmosphere in the ceramic crucible with and without a cover relative to the pure filler without treatment is compared. Further, the images of the fillers out of the scanning electron microscope are portrayed. The sample after incineration in the crucible without the cover shows a significant coating of the particle, which cannot be observed with the cover. Further, the composition of the elements reveals an increase of oxygen for the sample

without the cover and a reduction, especially of the ferrite (Fe) and neodymium (Nd) conjunction, relative to the particle before incineration. The sample with the cover even demonstrates an increase of Fe and Nd individually and in conjunction.

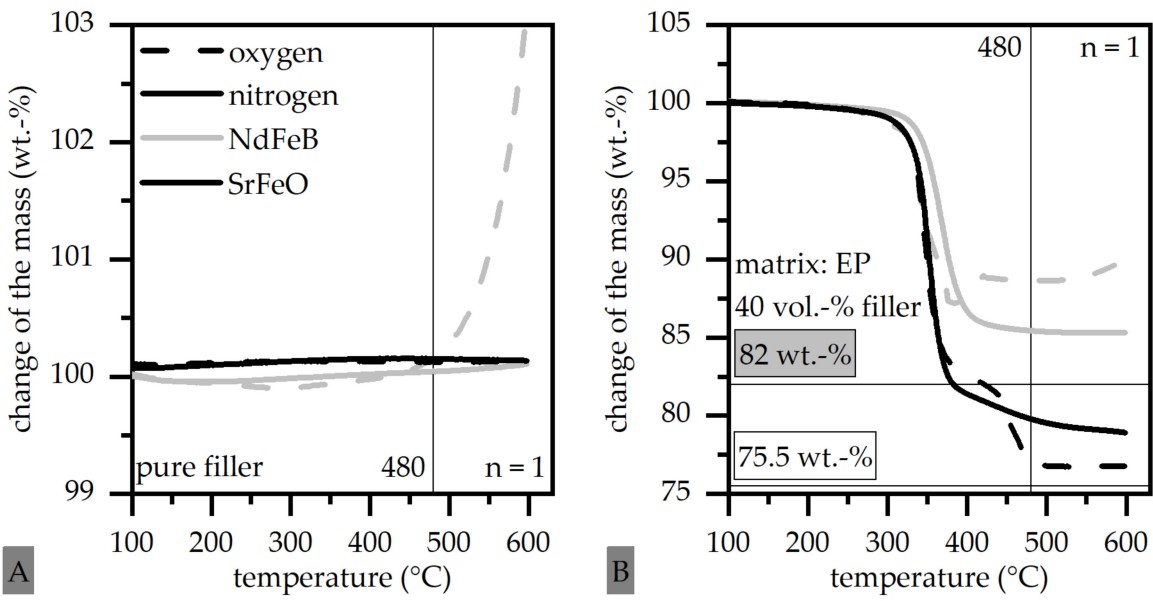

**Figure 10.** Change of the mass of pure filler (**A**) and compound with 40 vol.-% (**B**) of NdFeB and SrFeO with oxygen or nitrogen atmosphere.

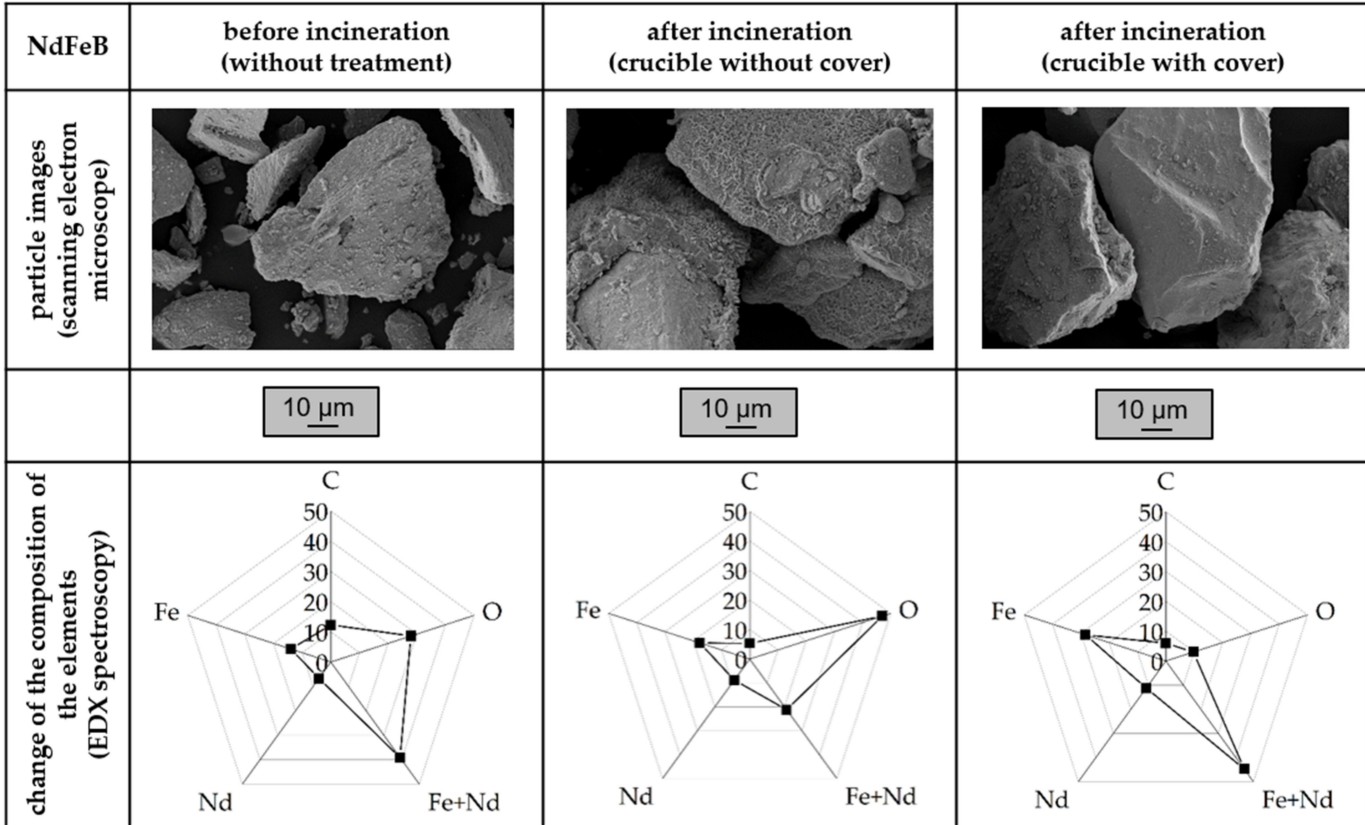

**Figure 11.** Change of the composition of the elements in the pure filler NdFeB with oxygen atmosphere during incineration in a crucible with and without cover compared to sample without treatment (EDX spectroscopy and scanning electron microscope).

Both analyses are further shown for SrFeO in Figure 12. Here, a change in the surface of the samples is not portrayed. A possible oxygen cover cannot be seen. However, the change of the composition of the elements reveals an increase of oxygen for the sample after the incineration in a crucible without the cover, but no change in the Fe or Strontium (Sr) concentration compared to the particles without treatment. Fe is significantly increased in the sample with cover.

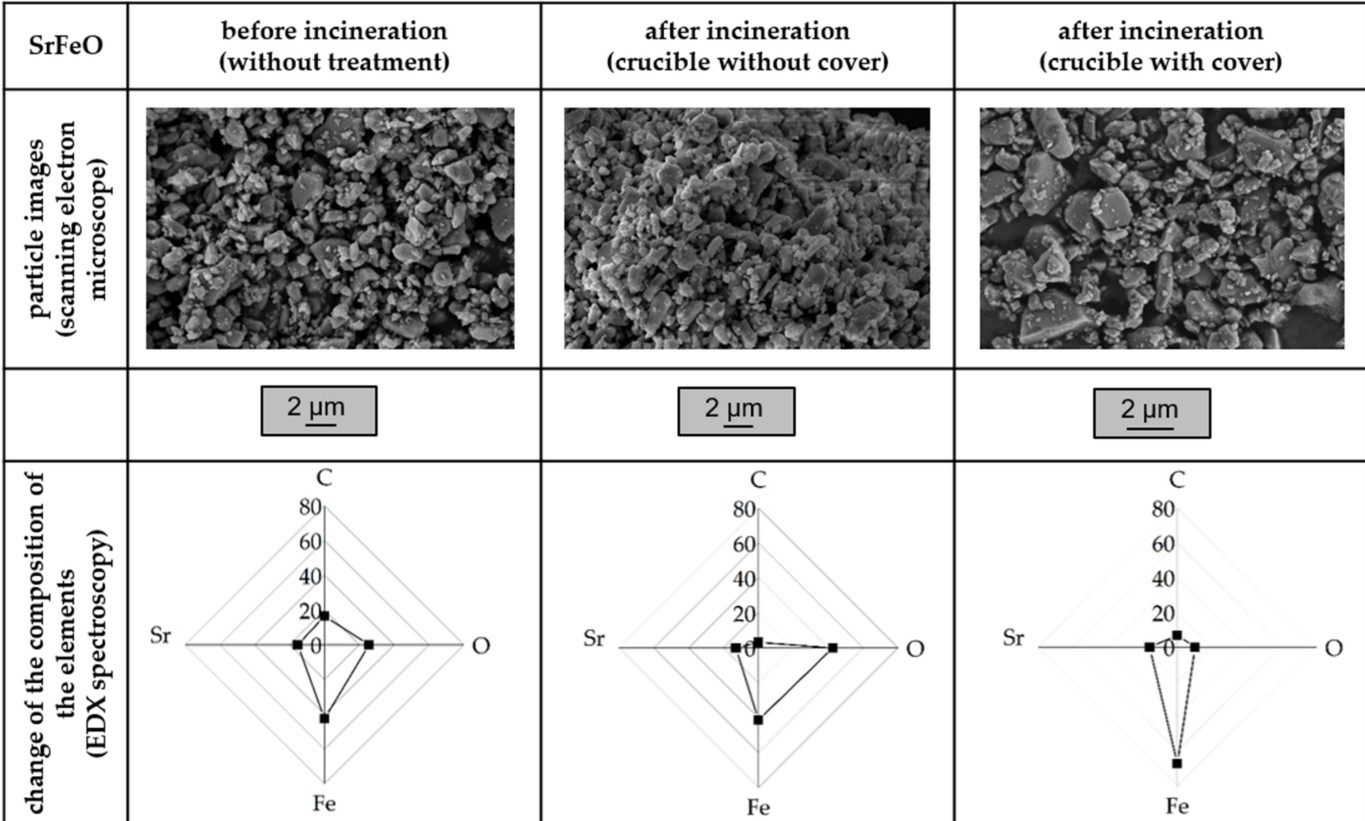

**Figure 12.** Change of the composition of the elements in the pure filler SrFeO with oxygen atmosphere during incineration in a crucible with and without cover compared to sample without treatment (EDX spectroscopy and scanning electron microscope).

Due to the heat treatment, in terms of the incineration Fe and in terms of NdFeB, Nd is concentrated in the outer areas of the filler, which leads to a reduction of these elements in the core. If an oxygen atmosphere is present during the incineration, further oxygen is built on the outer areas. With that, the higher quantity of elements supporting the magnetic properties cannot be used. The change of the concentration of the elements Fe and Nd in the outer and core regions do not affect the magnetic properties. They are influenced by the oxygen cover.

### 3.3. Magnetic Properties of Thermoplastic-Based Polymer-Bonded Magnets with Respect to the Mechanical Reuse of Shreds

The magnetic flux density was reduced by about 20% due to the recycling treatment of the mechanical reuse of shreds. The change in the magnetic properties of the recycled samples is shown in Figure 13A. Further, the results of the DSC measurement on the thermoplastic compound and the reused shreds are portrayed in Figure 13B. The route of the cooling behavior is shifted to lower temperatures compared to the thermoplastic compound and a double peak can be seen in the heating behavior.

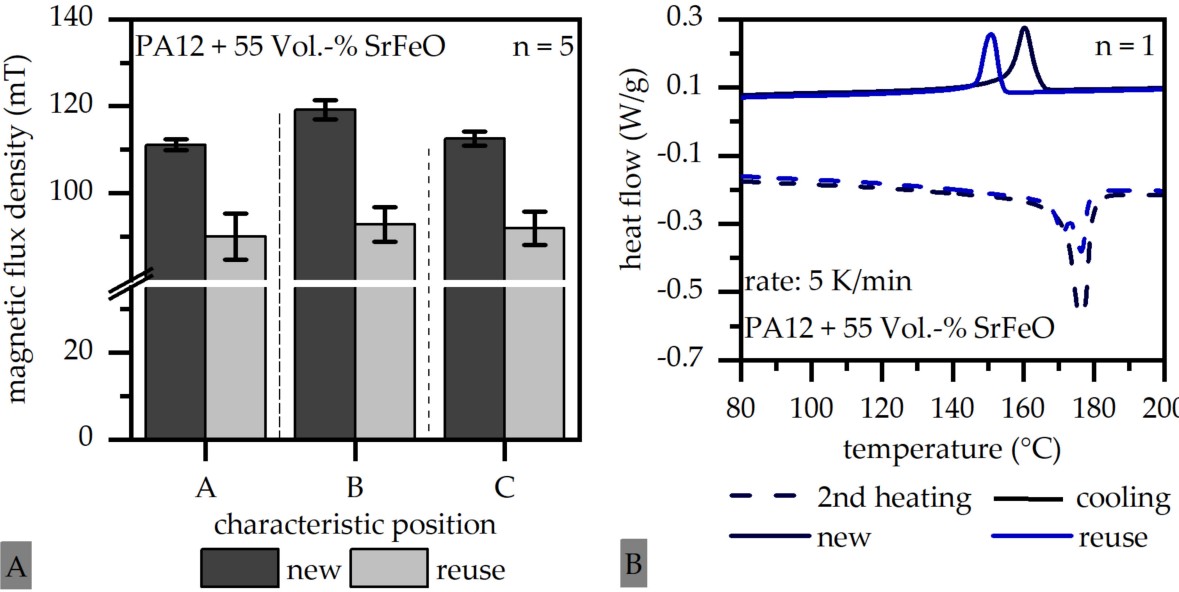

**Figure 13.** Change of the magnetic flux density due to reuse of shreds (**A**) and DSC measurement (**B**) based on thermoplastic compound PA12 with 55 vol.-% SrFeO.

The material of the reused shreds is fused a third time, if recycled samples should be fabricated. This leads to an aging process in the matrix material. Due to the thermal degradation, polymer chains are broken, and post-cross-linking processes take place. Both mechanisms led to a change in the processing parameters to guarantee the fabrication of the material. The DSC measurements revealed a thermal degradation due to the shift of the cooling behavior and the double peak during heating, which revealed a crystallization at low temperatures and a further recrystallization at the second peak [31]. In addition to this change, the reason for the reduction of the magnetic properties in general was found by analyzing the filler orientation and placement in the sample. Comparing the microstructure by the thermoplastic compound and the reused shred by a scanning electron microscope, a local concentration of matrix or filler and air gaps can be observed as shown in Figure 14. Further, the orientation of the samples is shown for the middle and the edge area of the sample in the middle of the pole and the pole crossover.

Within the thermoplastic compound, the poles broaden in the edge zone relative to the pole crossover and disorientate in the edge zone at the middle of the pole. Ref. [32] determined a fast-cooling edge zone, which led to the shift of the poles contrary to the flow direction. Samples fabricated out of the mechanically used shreds revealed a reduction of the pole width relative to the pole crossover without disorientation in the edge zone. The shift of the poles was parallel to the actual flow direction. It was assumed that the flow conditions of the shreds were different compared from the ones of the thermoplastic compound. With respect to the DSC measurement (compare with Figure 13B), the aging process based on the thermal degradation led to broken polymer chains and to shorter chains. Those chains reduced the viscosity of the material and increased the possibility of proper orientation, even in the edge zone. Further, the shift of the cooling behavior towards lower temperatures gave more time for the orientation due to a slower solidification. Therefore, the change in the molecular setting of the material improved the conditions for a uniform orientation and with that, increased magnetic properties. However, the magnetic flux density was reduced in terms of local concentrations and inhomogeneity within the material as well as air gaps. Compared to the thermoplastic compound, the peak position of the magnetic flux density was shifted, which led to a reduced accuracy, but the maximal value of the peak was similarly independent of the position. The change in the pole width is shown in Figure 15 for the thermoplastic compound and the mechanically used shreds to illustrate the difference in the flow conditions of both materials. In Figure 15 capital letters

represent the characteristic poles and the middle of the pole and lower cases correspond to the pole crossover. The gating point is placed at B and the welding line at a and d. The position of the welding line is not changed due to the different material, but the pole width in between the welding lines is influenced.

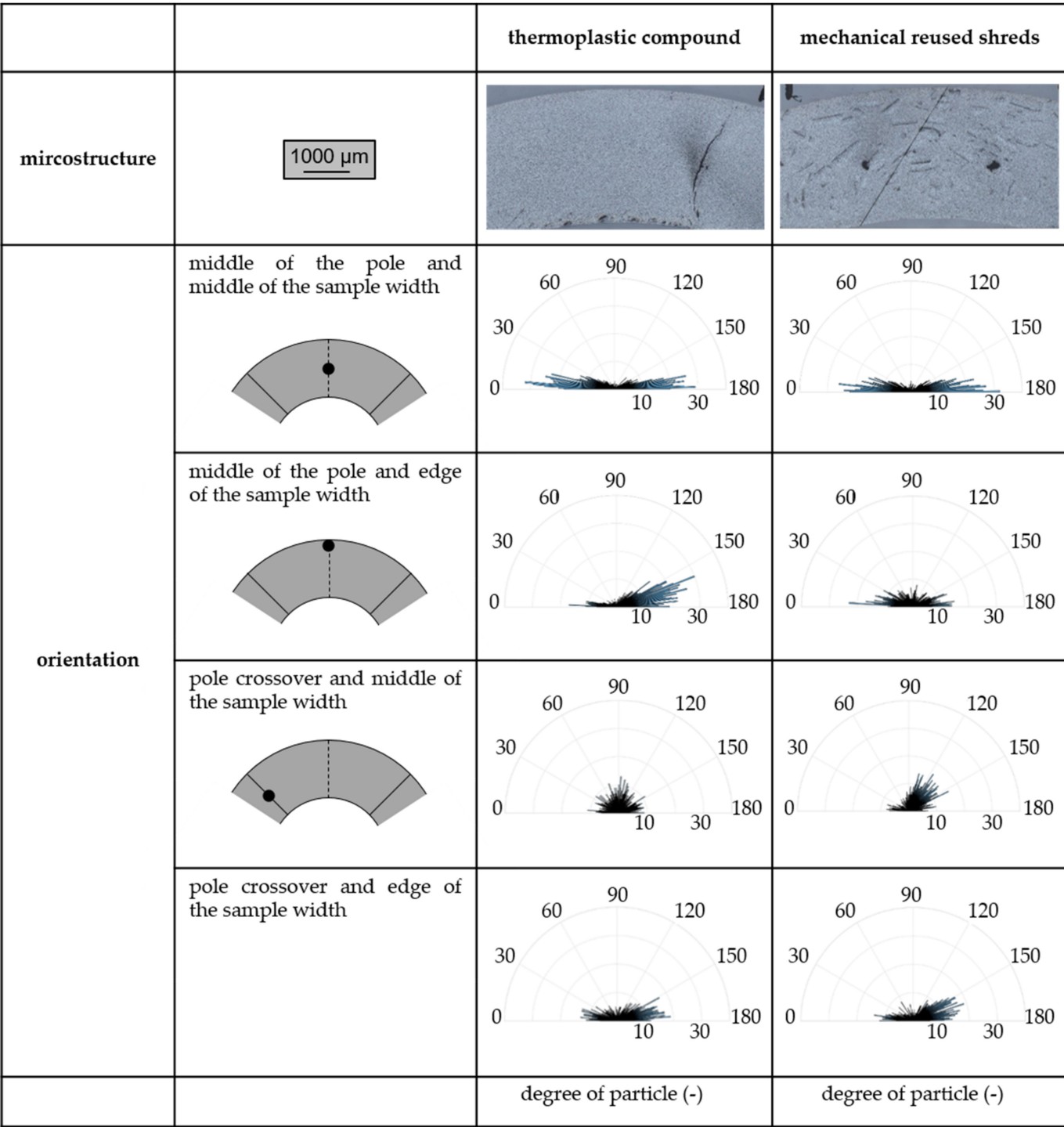

**Figure 14.** Comparison of the microstructure and the orientation at different positions in the sample relative to the pole crossover or middle and to the edge or middle of the sample width for thermoplastic compound and mechanically reused shreds.

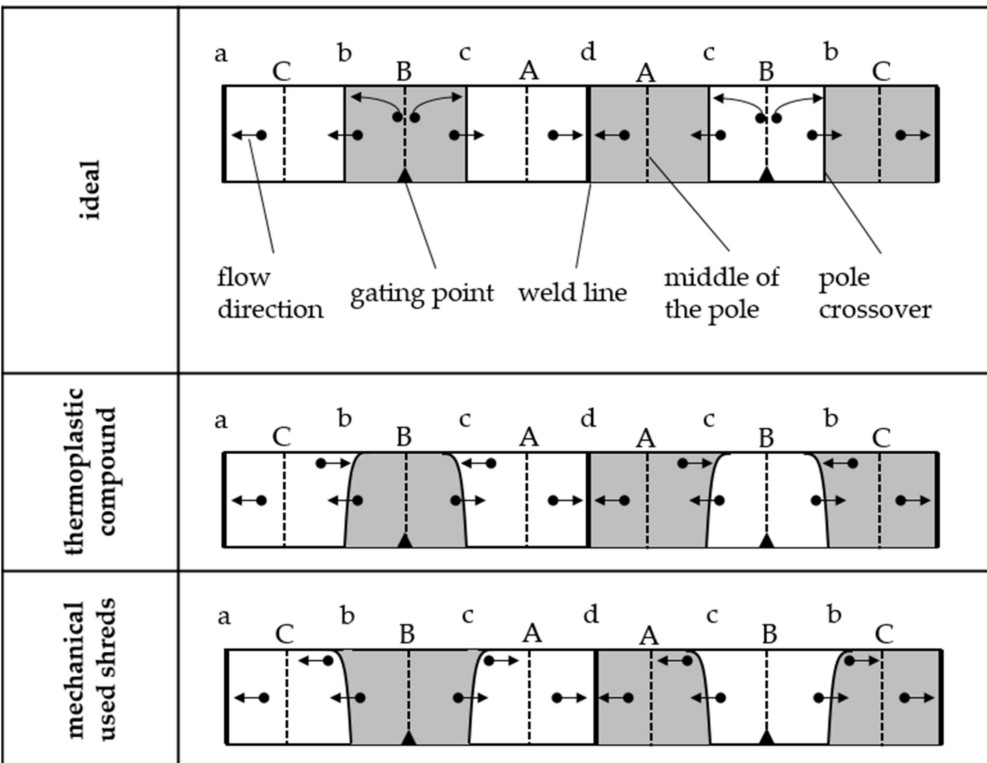

**Figure 15.** Change of the pole width relative to the flow direction and the gating point for thermoplastic compound and mechanically used shreds [A/B/C: characteristic poles and middle of the poles; a/b/c/d: pole crossover].

## 4. Conclusions

It was successfully shown that two different recycling strategies could be used for polymer-bonded magnets relative to the matrix system. In terms of the thermal recycling strategy for thermoset-based polymer-bonded magnets, the recovery of the filler material independently from the matrix material system was proven. The atmosphere during incineration and the process management determined the oxidation of the fillers and the change in the magnetic properties. The incineration further led to a shift of the filler elements without influencing the magnetic properties. Similar magnetic properties can be reached using the thermal strategy with the right process management.

In terms of the mechanically reused shreds based on a thermoplastic compound, the magnetic properties were reduced by about 20%. This could be reduced by adding new material partly to the shreds. Further, the flow conditions and the filler orientation were changed due to thermal degradation. However, both recycling strategies could successfully be used for polymer-bonded magnets, leading to the first options for recycling polymer-bonded magnets directly.

Future attempts will analyze the usage of the strategies for postconsumer materials, where the sample dimension as well as further foreign substances change the conditions of the recycling process. Additionally, the strategies should be assessed economically, taking the demagnetization into account.

**Author Contributions:** U.R.: conceptualization, methodology, validation, investigation, writing—original draft, visualization; D.D.: writing—review and editing, supervision, project. All authors have read and agreed to the published version of the manuscript.

**Funding:** We acknowledge financial support by Deutsche Forschungsgemeinschaft and Friedrich-Alexander-Universität Erlangen-Nürnberg within the funding program "Open Access Publication Funding".

**Data Availability Statement:** Restrictions apply for the availability of these data. Data are available with the permission of the author.

**Conflicts of Interest:** The authors declare no conflict of interest. The funders had no role in the design of the study; in the collection, analyses, or interpretation of data; in the writing of the manuscript, or in the decision to publish the results.

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
