# Peer review of "Possibilities in Recycling Magnetic Materials in Applications of Polymer-Bonded Magnets"

_2673-8724, doi:10.3390/magnetism2030019_

Round 1

Reviewer 1 Report

No comments

Author Response

Dear Sir or Madam, 

thank you very much for your kind reply and the time for reviewing my article. 

Kind regards

Uta Rösel 

Reviewer 2 Report

The comments were taken into consideration and the paper was corrected accordingly.

Author Response

(The authors gave the same response as above.)

Reviewer 3 Report

The response provided by the authors is not satisfactory on the basis of following grounds:

  1. The provided introduction is very exhaustive and covers the topic in a very broad sense and at several locations in an ambiguous way. The research design, experiments and results are focussed only on a part of the introduction and largely does not discuss the topics being focussed in the introduction. I suggest authors that kindly summarize the Introduction in a concise and focussed way for the ease of understanding among the readers. For instance, the article largely covers the macroscopic properties of the materials of interest but on Page 2 in Line 63-76 discusses the origin of magnetic moment. It is good to make the introduction detailed but such details can bring overexpectations among readers for the subsequent discussion on the studied materials from these perspectives as well.
  2. The use of word reduction/oxidation in chemistry is mostly related to the oxidation state of the material. On page 13, Line 471-479, authors mention that: "In the oxygen atmosphere further reduction takes place as agglomerates are built for SrFeO and an orientation in flow direction occurs for NdFeB." Kindly explain what are the chemical changes that oxygen is creating in the system that authors have referred as reduction? I suggest authors to pursue the X-ray photoelectron spectroscopic analysis any other appropriate technique to understand the change in oxidation state of an element/composition which could be useful for explaining the observations.

In response to this question, authors mention that: “The oxidation of hard magnetic fillers is a large field of research that should not and cannot be considered in detail in the paper. The authors wanted to prove the oxidation on the particles; an analysis of the fundamentals of oxidation is the basis for further comprehensive investigations.”

In my opinion, in this case the choice of wordings should either be changed or use appropriate reference to support their claim. Writing a sentence without any scientific evidence/reference may be challenging to convince the readership of the journal.

  1. In connection to the another question: Kindly unify the Figures in the article. For example, for SEM images, the way of representation of scale in Figure 2 and Figure 10 are very different and may cause an inconvenience for the interested readers.

The author response is: The authors are aware that the microscopic images have different scales. In the context of the paper, two fillers with significantly different filler sizes were considered. Therefore, these can only be illustrated in different scales, to visualize the effects in a appropriated way

I thank the authors for the clarification. However, choosing the particles of different sizes and of different chemical composition leads to the variation of two parameters at a time for a given measurement. Therefore, making a judgment for the observed results could be because of either or both of the variations in the used material.

I suggest authors kindly mention the reason for choosing the particles of such wide variation in size and include the particle size distribution in the manuscript.

Author Response

Dear Sir or Madam, 

thank you very much for your reply and the time for reviewing my article. Please check the response attached.

Kind regards

Uta Rösel 

Reviewer 4 Report

The manuscript "Possibilities in recycling magnetic materials..." by Rosel and Drummer provides a very thorough review of polymer bonded magnet recycling and presents data on two methods for recycling PBMs. The manuscript is high quality and ready for publication. The presentation is clear and the results point to the thermal method as a viable strategy for PBM recycling (shredding appears to be limited in terms of multiple recycling cycles). While the introduction is, in my opinion, a bit longwinded, it does provide a thorough review of the field. This manuscript is ready for publication.

Author Response

(The authors gave the same response as above.)

Round 2

Reviewer 3 Report

Authors have incorporated the recommended suggestions.

This manuscript is a resubmission of an earlier submission. The following is a list of the peer review reports and author responses from that submission.

Round 1

Reviewer 1 Report

The article aims to study the recycling methods for magnetic materials, i.e., polymer bonded magnets. However, the manuscript is not recommended to be accepted.

 (1) The innovation of this investigation is very limited, and new findings and exciting results are few. (2) Writing and English are poor and must be carefully improved. e.g., “Polymer bonded magnets increase significantly in the application of drive technology or more precise in terms of new concepts for the magnetic excitation of synchronous or direct current 8 (DC) machines”, ”Further the usage of recycled material shows a change in the flow 20 behaviour and with that an influence on the pole accuracy”, etc. (3) It is very confusing to read “The magnetic properties are reduced about 20 % which can be faced by using new material partly”? (4) Some information, i.e., thermoplastic and thermoset in Figure 1 are contradicted with descriptions with those in abstract (there should be something wrong in the abstract). (5) In Figure 8, why not present the microscopic photos with the same scale? How to measure or evaluate the orientation in detail? 

Reviewer 2 Report

The paper is well written, and it represents a novelty in the field of recycling magnetic composites. The paper is well structured with information about the state of the art, methodology, results, and the conclusions. There are minor changes that should be corrected to improve the manuscript.
1. I think that introduction section could be shortened since it is quite extended. 
2. The results of DSC measurements are very descriptive. Can the authors explain why there are shifts in the peaks? 
3. In the abstract the authors claim that the results show a change in the flow behavior, however, there are no results presented about the flow behavior of the materials used. The flow behavior is again mentioned in the conclusion section. Can the authors explain how they determined the change in flow behavior?
4. The authors should explain how they determined thermal degradation.
5. There are some typing errors, like:
315 - "...the samples was placed ..." it should be "... the samples were placed ..."
360 - ".. the ring were picked up ... " it should be "... the rings were picked up ...."
Figure caption 7: it should be filler instead of fuller
559 - "Both mechanism lead to a change ..." it should be "Both mechanisms led to a change ..."

Reviewer 3 Report

The article describes two methods of recycling anisotropic magnetic materials for polymer bonding materials. The article is comprehensively written in terms of language and content. However, there are multiple scopes of improvement in the manuscript, which are suggested to be incorporated for the better understanding. The suggestions/comments are as follows:

  1. The article is written in a form of laboratory report/thesis. I suggest authors to follow the journal guidelines.
  2. The introduction of the manuscript is too ambiguous and stresses on the topics/content which are either not directly relevant to the manuscript or/and quite difficult for general readers of the journal. For instance, on page 3, Line 115 "...research project FKZ 03VNE2052D...". I suggest authors to kindly remove unnecessary information from the article or kindly explain if it is necessary.
  3. Kindly unify the Figures in the article. For example, for SEM images, the way of representation of scale in Figure 2 and Figure 10 are very different and may cause an inconvenience for the interested readers. 
  4. In the experimental section, kindly explain the characterization conditions rather than working principle (for instance section 2.4.4). Unless the used technique is highly customized, the working principle of the technique may not be required.  
  5. In the results and discussion section, I suggest authors to include the standard deviation among the various plots/data to understand the reproducibility and/or extent of error being observed in these experiments. 
  6. On page 13, Line 465-470, authors mention that: "In the oxygen atmosphere further reduction takes place as agglomerates are built for SrFeO and an orientation in flow direction occurs for NdFeB." Kindly explain what are the chemical changes that oxygen is creating in the system that authors have referred as reduction? I suggest authors to pursue the X-ray photoelectron spectroscopic analysis any other appropriate technique to understand the change in oxidation state of an element/composition which could be useful for explaining the observations. 
  7. Authors focused on incineration atmosphere for playing a detrimental role of the fillers and the magnetic properties. However, the correlation between extent of oxidation and magnetic properties is not very clear. Kindly explain the research novelty clearly.